 **eLIFE**

# De novo synthesis of a sunscreen compound in vertebrates

**Andrew R Osborn[1†], Khaled H Almabruk[1†], Garrett Holzwarth[2,3†], Shumpei Asamizu[1‡], Jane LaDu[4], Kelsey M Kean[5], P Andrew Karplus[5], Robert L Tanguay[4], Alan T Bakalinsky[2], Taifo Mahmud[1]\***

[1]Department of Pharmaceutical Sciences, Oregon State University, Corvallis, United States; [2]Department of Food Science and Technology, Oregon State University, Corvallis, United States; [3]Department of Microbiology, Oregon State University, Corvallis, United States; [4]Department of Environmental and Molecular Toxicology, Oregon State University, Corvallis, United States; [5]Department of Biochemistry and Biophysics, Oregon State University, Corvallis, United States

**\*For correspondence:** taifo. mahmud@oregonstate.edu

[†]These authors contributed equally to this work

**Present address:** [‡]Department of Biotechnology, Graduate School of Agricultural and Life Sciences, The University of Tokyo, Bunkyo, Japan

**Competing interests:** The authors declare that no competing interests exist.

**Abstract** Ultraviolet-protective compounds, such as mycosporine-like amino acids (MAAs) and related gadusols produced by some bacteria, fungi, algae, and marine invertebrates, are critical for the survival of reef-building corals and other marine organisms exposed to high-solar irradiance. These compounds have also been found in marine fish, where their accumulation is thought to be of dietary or symbiont origin. In this study, we report the unexpected discovery that fish can synthesize gadusol de novo and that the analogous pathways are also present in amphibians, reptiles, and birds. Furthermore, we demonstrate that engineered yeast containing the fish genes can produce and secrete gadusol. The discovery of the gadusol pathway in vertebrates provides a platform for understanding its role in these animals, and the possibility of engineering yeast to efficiently produce a natural sunscreen and antioxidant presents an avenue for its large-scale production for possible use in pharmaceuticals and cosmetics.

## Introduction

The sunscreen compounds, mycosporine-like amino acids (MAAs) and related gadusols, commonly found in bacteria, fungi, algae, and marine invertebrates (*Shick and Dunlap, 2002*; *Miyamoto et al., 2014*), have been proposed to fulfill a variety of functions, such as sunscreen, antioxidant, stress response, intracellular nitrogen reservoir, and/or optical filter (*Gao and Garcia-Pichel, 2011*; *Bok et al., 2014*). Although their formation had long been proposed to originate from the shikimate pathway, more recent bioinformatic and biochemical studies revealed that in cyanobacteria, MAAs are synthesized by desmethyl-4-deoxygadusol synthase (DDGS), a dehydroquinate synthase (DHQS) homolog (*Wu et al., 2007*; *Balskus and Walsh, 2010*; *Singh et al., 2010*; *Asamizu et al., 2012*). Interestingly, inactivation of the DDGS gene in *Anabaena variabilis* ATCC 29413 did not abolish the production of MAAs, suggesting that additional pathways exist for the biosynthesis of MAAs (*Spence et al., 2012*). DDGS converts sedoheptulose 7-phosphate (SH7P) to desmethyl-4-deoxygadusol via a unique sequence of dephosphorylation, aldol condensation, enolization, dehydration, reduction, and tautomerization reactions (*Figure 1—figure supplement 1*) (*Balskus and Walsh, 2010*; *Asamizu et al., 2012*). The product is subsequently converted by a methyltransferase to 4-deoxygadusol, the building block of MAAs.

4-Deoxygadusol has also been proposed to be the precursor of gadusol (*Starcevic et al., 2010*; *Rosic and Dove, 2011*), a related compound initially isolated from cod roe (*Gadus morhua* L.) (*Plack et al., 1981*), but also found in roes of other marine fish (*Plack et al., 1981*; *Arbeloa et al., 2010*),

**eLife digest** Sunlight is the Earth's primary energy source and is exploited by an array of natural and man-made processes. Photosynthetic plants harness solar energy to convert carbon dioxide and water into biomass, and solar panels capture light and convert it to electricity. Sunlight is critical to life on Earth, and yet excessive exposure to sunlight can cause serious harm as it contains ultraviolet (UV) radiation, which damages the DNA of cells. In humans, this damage can lead to conditions such as cataracts and skin cancer.

The marine organisms and animals that live in the upper ocean and on reefs are subject to intense and unrelenting sunlight. In their effort to protect against potentially deadly UV radiation, many small and particularly vulnerable marine organisms, such as bacteria and algae, produce UV-protective sunscreens. While UV-protective compounds have also been found in larger organisms, including fish and their eggs, the presence of these sunscreens has always been attributed to the animal sequestering the compounds from their environment or partnering with a sunscreen-producing microorganism.

Now, Osborn, Almabruk, Holzwarth et al. have discovered a fish that is able to produce such a UV-protective compound completely on its own. After identifying the full set of genes—or pathway—responsible for generating the UV-protective compound, the same pathway was detected in a variety of diverse animals, including amphibians, reptiles, and birds. This opens up a new area of study, because besides providing UV protection, no one yet knows what other roles the molecule may have in these animals. Furthermore, introducing the complete pathway into yeast enabled these cells to produce the sunscreen. In the future, engineering a yeast population to produce large quantities of the natural sunscreen could lead to large-scale production of the UV-protective compound so it can be used in pharmaceuticals and cosmetics.

sea urchin eggs (*Chioccara et al., 1986*), cysts and nauplii of brine shrimp (*Grant et al., 1985*), mantis shrimp crystalline cones (*Bok et al., 2014*), and sponges (*Bandaranayake et al., 1997*). As genes responsible for the production of 4-deoxygadusol and MAAs are commonly found in bacteria (e.g., cyanobacteria), algae, and other marine microorganisms (*Shick and Dunlap, 2002*; *Miyamoto et al., 2014*), the accumulation of these compounds in marine animals has been proposed to be of dietary or symbiont origin (*Arbeloa et al., 2010*; *Gao and Garcia-Pichel, 2011*; *Loew, 2014*). On the other hand, a gene cluster like that in cyanobacteria apparently encoding a four-step DDGS-based pathway for converting SH7P to MAAs was discovered in the genomes of a coral (*Acropora digitifera*) and sea anemone (*Nematostella vectensis*), suggesting that these invertebrates can produce MAAs autonomously (*Rosic and Dove, 2011*; *Shinzato et al., 2011*).

DDGS is a member of the sugar phosphate cyclase (SPC) superfamily (*Wu et al., 2007*). In addition to DHQS, this superfamily includes four enzymes known for their roles in the biosynthesis of natural products with therapeutic application: 2-epi-5-epi-valiolone synthase (EEVS), 2-epi-valiolone synthase, aminoDHQS, and 2-deoxy-*scyllo*-inosose synthase (DOIS) (*Wu et al., 2007*; *Mahmud, 2009*; *Asamizu et al., 2012*; *Kang et al., 2012*). EEVS catalyzes the entry step to the biosynthesis of pseudosugar-containing natural products, such as the antidiabetic drug acarbose and the crop protectant validamycin A, and has so far only been identified and characterized in bacteria (*Mahmud, 2009*).

Recently, we surprisingly also found genes that encode EEVS-like proteins (annotated as 'PREDICTED: pentafunctional AROM polypeptide-like') in the genomes of fish, amphibians, reptiles, and birds (*Figure 1A*, *Figure 1—figure supplement 2*, *Table 1*, *Figure 1—source data 1*). This gene is located in a cluster with another functionally unknown gene (labeled as MT-Ox hereafter) and flanked by a suite of transcription factor genes encoding FRMD4B, MitF, MDFIC, and FoxP1 (*Figure 1B*). These transcription factors have been known to regulate the expression of genes with essential roles in cell differentiation, proliferation, and survival (*Levy et al., 2006*; *Rice et al., 2012*; *Garner et al., 2014*; *Zhao et al., 2015*). However, whether they have a direct functional relationship with the EEVS and MT-Ox genes remains an open question. Whereas humans and other mammals also have FRMD4B, MitF, MDFIC, and FoxP1 homologs, they lack the EEVS-like and the MT-Ox genes (*Figure 1B*). Therefore, the presence of the EEVS-like gene, previously only found in bacteria, in the genomes of fish, and other egg-laying vertebrates is puzzling and evolutionarily intriguing. Here, we

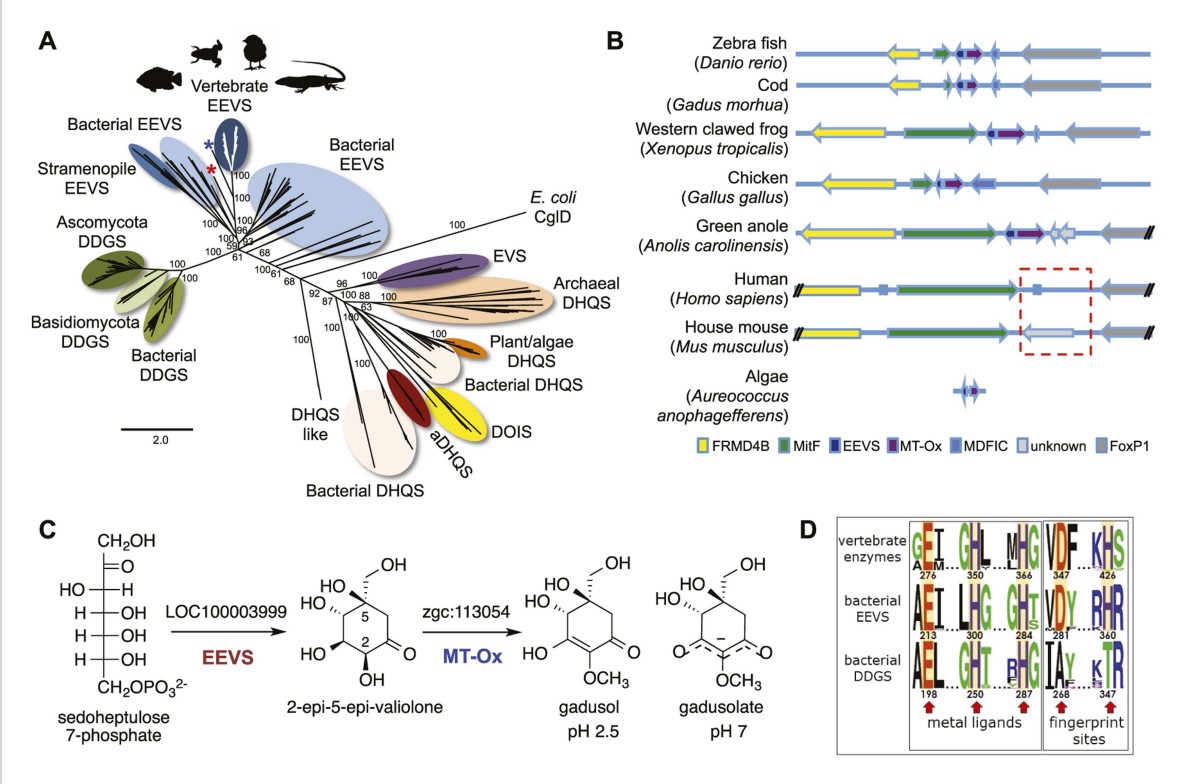

**Figure 1**. Bioinformatic analysis of sugar phosphate cyclases in prokaryotes and eukaryotes and biochemical characterization of gadusol biosynthetic enzymes. (**A**) Bayesian phylogenetic tree of SPCs. Numbers represent posterior probability. The stramenopile *Aureococcus anophagefferens*, denoted by the blue star, has EEVS and MT-Ox proteins strikingly similar (over 50% identical) to those in vertebrates. The micro algae *Coccomyxa subellipsoidea*, denoted by the red star, also has EEVS and MT-Ox. (**B**) Genetic organizations of EEVS and MT-Ox genes in fish, amphibians, birds, and reptiles. Humans and other mammals lack these genes (indicated in dashed red box). For a complete list of vertebrates whose genomes are known to contain EEVS and MT-Ox genes, see *Table 1*. FRMD4B, FERM domain-containing protein 4B; MitF, microphtalmia-associated transcription factor; MDFIC, MyoD-family inhibitor domain-containing protein-like; and FoxP1, Forkhead-related transcription factor 1. (**C**) Biochemical characterization of recombinant LOC100003999 and zgc:113054 proteins. (**D**) WebLogo (*Crooks et al., 2004*) images of residue conservation patterns at the three metal ligand positions and two active site fingerprint sites known (*Kean et al., 2014*) to distinguish bacterial EEVSs from DDGSs. The residue numbers given correspond to the reference proteins *D. rerio* EEVS, ValA, and *A. variabilis* DDGS, respectively. WebLogos were based on 126 vertebrate, 63 bacterial EEVS, and 160 bacterial DDGS sequences, respectively, that had BLASTP E-values <10$^{-120}$ in searches using the reference proteins noted above as queries. Each group was aligned using ProMals (*Pei and Grishin, 2007*).

The following source data and figure supplements are available for figure 1:

**Source data 1**. Amino acid sequences of the Sugar Phosphate Cyclases (SPCs).

**Figure supplement 1**. Biosynthetic pathway to shinorine in *Anabaena variabilis* (*Balskus and Walsh, 2010*).

**Figure supplement 2**. Maximum likelihood phylogenetic tree of sugar phosphate cyclases.

**Figure supplement 3**. SDS PAGE of EEVS and MT-Ox proteins and thin-layer chromatography (TLC) analysis of their products.

**Figure supplement 4**. GC-MS analysis of LOC100003999 and ValA reaction products.

**Figure supplement 5**. Sequences of the vertebrate clade of EEVS-related proteins have characteristic features of an EEVS.

**Figure supplement 6**. Stereoview for the modeled active site geometry of LOC100003999.

**Table 1**. Genetic analysis of EEVS and MT-Ox genes in some invertebrates and chordates

| Class | Species | Common name | FRMD4B | MitF | EEVS | MT-Ox | MDFIC | FoxP1 |
|---|---|---|---|---|---|---|---|---|
| Cnidaria | *Acropora digitifera* | Coral* | – | – | – | – | – | – |
| | *Nematostella vectensis* | Starlet sea anemone* | √ | √ | – | – | – | √ |
| | *Hydra vulgaris* | Common hydra | – | √ | – | – | – | √ |
| Invertebrate chordates | *Ciona intestinalis* | Vase tunicate | √ | √ | – | – | – | √ |
| | *Branchiostoma floridae* | Florida lancelet | √ | √ | – | – | – | √ |
| Agnatha | *Petromyzon marinus* | Lamprey† | ‡ | √ | ‡ | ‡ | ‡ | √ |
| Chondrichthyes | *Callorhinchus milii* | Australian ghostshark† | √ | √ | ‡ | ‡ | √ | √ |
| Osteichthyes (Actinopterygii) | *Astyanax mexicanus* | Mexican tetra | √ | √ | √ | √ | √ | √ |
| | *Danio rerio* | Zebra fish | √ | √ | √ | √ | √ | √ |
| | *Dicentrarchus labrax* | European seabass | √ | √ | √ | √ | √ | √ |
| | *Fundulus heteroclitus* | Mummichog | √ | √ | √ | √ | √ | √ |
| | *Gadus morhua* | Atlantic cod | √ | √ | √ | √ | √ | √ |
| | *Gasterosteus aculeatus* | Three-spined stickleback | √ | √ | √ | √ | √ | √ |
| | *Lepisosteus oculatus* | Spotted gar | √ | √ | √ | √ | √ | √ |
| | *Maylandia zebra* | Zebra mbuna | √ | √ | √ | √ | √ | √ |
| | *Neolamprologus brichardi* | African cichlid | √ | √ | √ | √ | √ | √ |
| | *Oncorhynchus mykiss* | Rainbow trout | √ | √ | √ | √ | √ | √ |
| | *Orechromis niloticus* | Nile tilapia | √ | √ | √ | √ | √ | √ |
| | *Oryzias latipes* | Japanese rice fish | √ | √ | √ | √ | √ | √ |
| | *Poecilia formosa* | Amazon molly | √ | √ | √ | √ | √ | √ |
| | *Pundamilia nyererei* | African cichlid | √ | √ | √ | √ | √ | √ |
| | *Salmo salar* | Atlantic salmon | √ | √ | √ | √ | √ | √ |
| | *Takifugu rubripes* | Japanese puffer | √ | √ | – | – | – | √ |
| | *Tetraodon nigroviridis* | Green spotted puffer | √ | √ | – | – | – | √ |
| | *Xiphophorus maculatus* | Southern platyfish | √ | √ | √ | √ | √ | √ |
| Sarcopterygii | *Latimeria chalumnae* | West African Coelacanth | √ | √ | – | – | √ | √ |
| Amphibia | *Xenopus tropicalis* | Western clawed frog | √ | √ | √ | √ | √ | √ |
| Reptilia | *Alligator mississippiensis* | American alligator | √ | √ | √ | √ | √ | √ |
| | *Anolis carolinensis* | Green anole | √ | √ | √ | √ | √ | √ |
| | *Chelonia mydas* | Green sea turtle | √ | √ | √ | √ | √ | √ |
| | *Chrysemys picta bellii* | Western painted turtle | √ | √ | √ | √ | √ | √ |
| | *Ophiophagus hannah* | King cobra† | √ | √ | ‡ | ‡ | √ | √ |
| | *Pelodiscus sinensis* | Chinese softshell turtle | √ | √ | √ | √ | √ | √ |
| | *Python bivittatus* | Burmese python† | √ | √ | ‡ | ‡ | √ | √ |
| Aves | *Anas platyrhynchos* | Mallard | √ | √ | √ | √ | √ | √ |
| | *Columba livia* | Rock dove | √ | √ | √ | √ | √ | √ |
| | *Falco cherrug* | Saker falcon | √ | √ | √ | √ | √ | √ |
| | *Ficedula albicollis* | Collard flycatcher | √ | √ | √ | √ | √ | √ |
| | *Gallus gallus* | Chicken | √ | √ | √ | √ | √ | √ |
| | *Geospiza fortis* | Medium ground-finch | √ | √ | √ | √ | √ | √ |
| | *Meleagris gallopavo* | North American wild turkey | √ | √ | √ | √ | √ | √ |
| | *Melopsittacus undulates* | Common pet parakeet | √ | √ | √ | √ | √ | √ |
| | *Pseudopodoces humilis* | Tibetan ground-tit | √ | √ | √ | √ | √ | √ |
| | *Taeniopygia guttata* | Zebra finch | √ | √ | √ | √ | ‡ | √ |
| | *Zonotrichia albicollis* | White-throated sparrow | √ | √ | √ | √ | √ | √ |

*Table 1. Continued on next page*

Table 1. Continued

| Class | Species | Common name | FRMD4B | MitF | EEVS | MT-Ox | MDFIC | FoxP1 |
|-------|---------|-------------|--------|------|------|-------|-------|-------|
| Mammalia | *Balaenoptera acutorostrata scammoni* | Minke whale | √ | √ | – | – | – | √ |
| | *Bos taurus* | Cattle | √ | √ | – | – | – | √ |
| | *Callithrix jacchus* | White-tufted-ear marmoset | √ | √ | – | – | – | √ |
| | *Capra hircus* | Goat | √ | √ | – | – | – | √ |
| | *Ceratotherium simum simum* | Southern white rhinoceros | √ | √ | – | – | – | √ |
| | *Chinchilla lanigera* | Long-tailed chinchilla | √ | √ | – | – | √ | √ |
| | *Chlorocebus sabaeus* | Green monkey | √ | √ | – | – | √ | √ |
| | *Cricetulus griseus* | Chinese hamster | √ | √ | – | – | √ | √ |
| | *Eptesicus fuscus* | Big brown bat | √ | √ | – | – | – | √ |
| | *Equus caballus* | Horse | √ | √ | – | – | – | √ |
| | *Erinaceus europaeus* | Western European hedgehog | √ | √ | – | – | √ | √ |
| | *Felis catus* | Cat | √ | √ | – | – | – | √ |
| | *Gorilla gorilla* | Western gorilla | √ | √ | – | – | – | √ |
| | *Heterocephalus glaber* | Naked mole-rat | √ | √ | – | – | √ | √ |
| | *Homo sapiens* | Human | √ | √ | – | – | – | √ |
| | *Lipotes vexillifer* | Yangtze river dolphin | √ | √ | – | – | – | √ |
| | *Loxodonta africana* | African savanna elephant | √ | √ | – | – | – | √ |
| | *Macaca fascicularis* | Crab-eating macaque | √ | √ | – | – | √ | √ |
| | *Mesocricetus auratus* | Golden hamster | √ | √ | – | – | √ | √ |
| | *Monodelphis domestica* | Gray short-tailed opossum | √ | √ | – | – | √ | √ |
| | *Mus musculus* | House mouse | √ | √ | – | – | √ | √ |
| | *Mustela putorius furo* | Domestic Ferret | √ | √ | – | – | – | √ |
| | *Myotis davidii* | Mouse-eared bat | √ | √ | – | – | – | √ |
| | *Myotis lucifugus* | Little-brown bat | √ | √ | – | – | √ | √ |
| | *Nomascus leucogenys* | Northern white-cheeked gibbon | √ | √ | – | – | – | √ |
| | *Odobenus rosmarus divergens* | Pacific walrus | √ | √ | – | – | √ | √ |
| | *Orcinus orca* | Killer whale | √ | √ | – | – | – | √ |
| | *Ornithorhynchus anatinus* | Duck-billed platypus† | √ | √ | – | – | – | √ |
| | *Orycteropus afer* | Ardvark | √ | √ | – | – | – | √ |
| | *Otolemur garnettii* | Small-eared galago | √ | √ | – | – | – | √ |
| | *Ovis aries* | Sheep | √ | √ | – | – | – | √ |
| | *Pan troglodytes* | Chimpanzee | √ | √ | – | – | – | √ |
| | *Papio anubis* | Olive baboon | √ | √ | – | – | – | √ |
| | *Peromyscus maniculatus bairdii* | Prairie deer mouse | √ | √ | – | – | √ | √ |
| | *Pteropus alecto* | Black flying fox | √ | √ | – | – | – | √ |
| | *Rattus norvegicus* | Norway rat | √ | √ | – | – | – | √ |
| | *Sarcophilus harrisii* | Tasmanian devil | √ | √ | – | – | √ | √ |
| | *Trichechus manatus latirostris* | Florida manatee | √ | √ | – | – | – | √ |
| | *Vicugna pacos* | Alpaca | √ | √ | – | – | – | √ |

*Harbors genes for MAA biosynthesis.

†Incomplete genome sequence; no EEVS and MT-Ox genes identified.

‡The presence of EEVS or MT-Ox genes is unknown due to missing contigs or sequence information.

report bioinformatics and functional studies of zebrafish (*Danio rerio*) EEVS-like and MT-Ox genes and the expression of this two-enzyme pathway in yeast, and discuss the evolutionary origin of this pathway in fish, amphibians, reptiles, and birds.

## Results and discussion

To investigate the function of the vertebrate EEVS-like genes, the protein encoded by the zebrafish EEVS-like gene (LOC100003999) was expressed in *Escherichia coli*. Incubation of the recombinant protein with SH7P gave a product, which was confirmed by thin-layer chromatography (TLC) and GC-MS to be EEV (*Figure 1C*, *Figure 1—figure supplements 3, 4*), identifying the protein as an EEVS. The best-characterized bacterial EEVS is ValA from the validamycin pathway in *Streptomyces hygroscopicus* subsp. *jinggangensis* 5008 (*Bai et al., 2006*), and the crystal structure of ValA (Protein Data Bank [PDB] entry 4P53) (*Kean et al., 2014*), allowed identification of a fingerprint set of 14 active-site residues with characteristic variations that could differentiate the various SH7P cyclases. Further supporting the assignment of the LOC100003999-encoded protein as an EEVS, sequence comparisons show that all animal EEVS-like proteins are highly similar (60–72% identity) and also match the sequence of ValA at all 14 fingerprint sites (*Figure 1D*, *Figure 1—figure supplements 5, 6*). This firmly establishes the presence of EEVS activity in animals.

The second gene, MT-Ox (zgc:113054), is predicted to encode a protein that contains two domains: the *N*-terminal domain is similar to *S*-adenosylmethionine (SAM)–dependent methyltrans-ferases, and the *C*-terminal domain is similar to $NAD^+$-dependent oxidoreductases. Although its function in zebrafish is unknown, the transcription of this gene in larvae is upregulated by light, leading to a prediction of its involvement in circadian clock regulation (*Weger et al., 2011*). In contrast, we hypothesized that this bifunctional protein is involved in modifying EEV to yield an oxidized and methylated product (*Figure 1C*). To test this hypothesis, recombinant MT-Ox protein encoded by zgc:113054 was incubated with EEV in the presence of SAM and $NAD^+$. Following incubation, a product with λmax of 294 nm (pH 7) and 270 (pH 2.5) was detected (*Figure 2A–B*). Further analysis of the product by (−)-ESI-MS (*m/z* 203 [M-H]$^-$) and $^1$H NMR confirmed its identity as gadusol (*Figure 2—figure supplements 1, 2*). We postulate that the conversion of EEV to gadusol by MT-Ox takes place via oxidation of the C-2 or C-3 OH, followed by enolization and methylation of the resulting C-2 OH (*Figure 2—figure supplement 3*). This new pathway to the UV-absorbing vinylogous acid functional group shared by gadusol and 4-deoxygadusol is distinct from that used in the biosynthesis of the MAAs (*Balskus and Walsh, 2010*; *Spence et al., 2012*), and the existence of multiple pathways for generating this scaffold is intriguing and may indicate that this is a privileged chemical scaffold in living organisms.

In zebrafish, both of the LOC100003999 and zgc:113054 genes are expressed during embryonic development. qRT-PCR analysis of mRNA isolated from zebrafish embryos at 12, 24, 48, 72, 96, and 120 hpf showed maximal expression at 72 hpf (*Figure 2C–D*). To demonstrate de novo synthesis of gadusol in zebrafish, the embryos were collected at 72 hpf, lyophilized and extracted with methanol, and the extract was analyzed by HPLC and ESI-MS (*Figure 2G*). Our finding of gadusol in the extract unambiguously confirms the ability of zebrafish to synthesize gadusol and amends the current perception that gadusol found in fish and other vertebrates is necessarily of dietary or symbiont origin. However, as MAAs are synthesized via a different pathway and there is no evidence that fish have those biosynthetic enzymes, the accumulation of MAAs in fish would still appear to be of dietary origin (*Mason et al., 1998*; *Zamzow, 2004*).

To show that the recombinant LOC100003999 and zgc:113054 genes are sufficient for encoding gadusol synthesis, they were cloned into a yeast expression vector and transferred into a *Saccharomyces cerevisiae* strain, in which the transaldolase gene *TAL1* had been deleted. Yeast possesses a robust pentose–phosphate pathway (*Figure 2—figure supplement 4*), and by removing the transaldolase enzyme, which normally metabolizes SH7P, and adding EEVS and MT-Ox, we expected to facilitate an effective shunt pathway from SH7P to gadusol. Analysis of the culture broth by HPLC, ESI-MS, and UV spectrophotometry revealed the presence of gadusol (*Figure 2H*) and its accumulation to ~20 mg/l after 5 days (*Figure 2E*). The results not only demonstrate the ability of the engineered yeast to produce and secrete gadusol but also present a new avenue for large-scale production of the compound for possible commercial uses, for example, sunscreen and/or antioxidant (*Plack et al., 1981*; *Schmid et al., 2006*; *Cardozo et al., 2007*; *Arbeloa et al., 2010*).

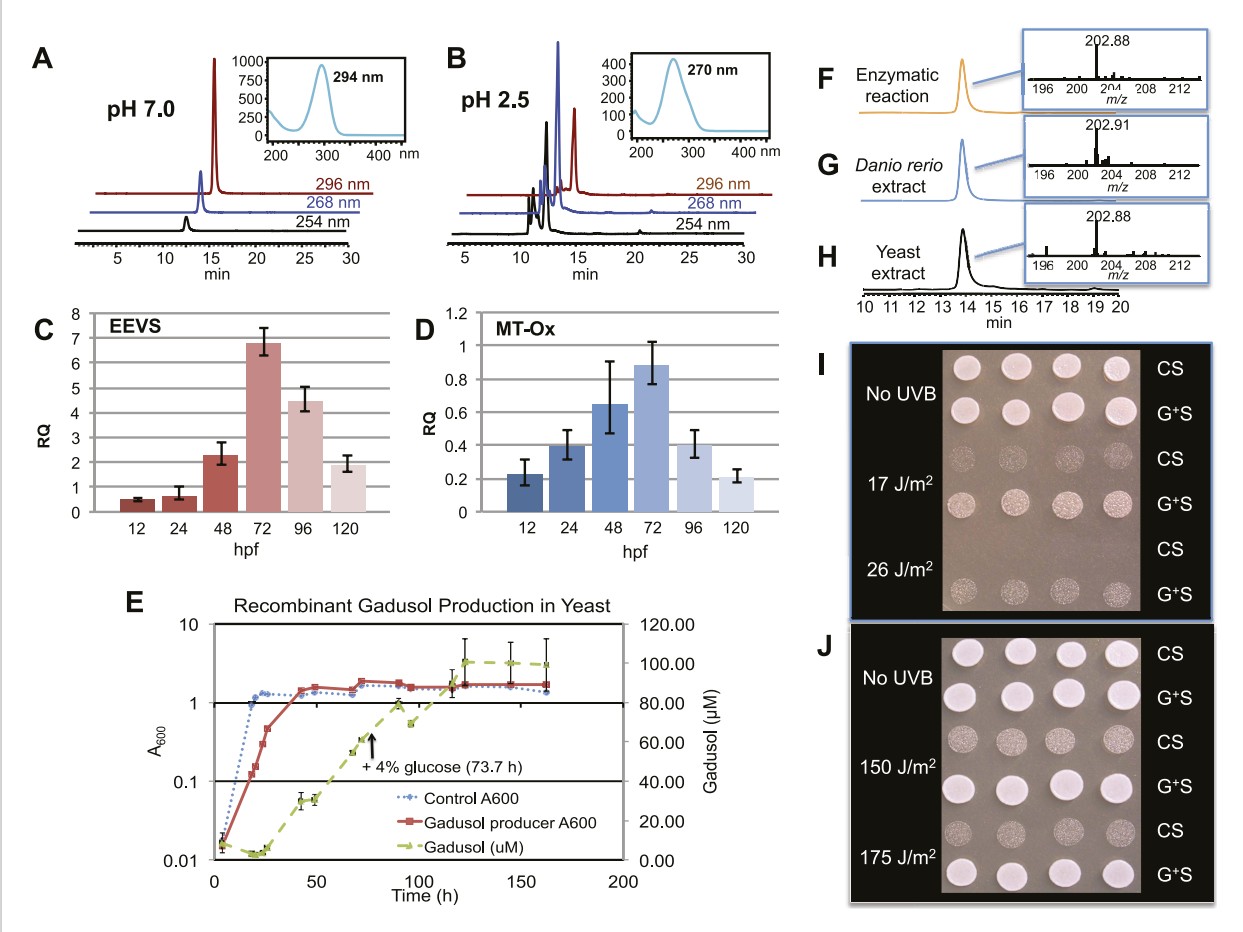

**Figure 2**. Production of gadusol in zebrafish and yeast and its sunscreen activity. (**A–B**) HPLC traces and UV absorptions of gadusol produced from *Escherichia coli* cell-free extract containing EEVS and purified MT-Ox protein at pH 7.0 and 2.5. (**C–D**) Transcription patterns of EEVS and MT-Ox genes during zebrafish embryonic development. qRT-PCR analysis of mRNA isolated from zebrafish embryos (n = 3) at 12, 24, 48, 72, 96, and 120 hpf. (**E**) Time course of gadusol production in yeast harboring the zebrafish genes. The yeast was cultured in YNB + 2% glucose supplemented with leucine and lysine at 30°C for 2 days, and growth was monitored as $A_{600}$ values (control, dotted blue line; gadusol producer, solid red line). Gadusol concentration in the supernatant of 20 ml cultures (n = 3) was monitored as $A_{296}$ values in 50 mM phosphate buffer, pH 7.0 (dashed green line) corrected for non-gadusol background absorbance in the control supernatant, normalized to $A_{600}$ value. Gadusol was quantified based on an extinction coefficient of 21,800 $M^{-1}$ $cm^{-1}$ in 50 mM phosphate buffer, pH 7. (**F–H**) Comparative HPLC analysis of gadusol from recombinant enzymatic reaction, zebrafish extract, and yeast extract. (**I**) Gadusol suppresses the UVB sensitivity of a *rad1Δ* yeast mutant; and (**J**) Gadusol increases the UVB tolerance of a wild-type (*RAD1*) strain. Cells suspended in control supernatant (CS) or gadusol+ supernatant (G+S) were irradiated with UVB and subsequently spotted in 3 μl aliquots (n = 4) onto YEPD plates, which were incubated at 30°C for 24 hr.

The following figure supplements are available for figure 2:

**Figure supplement 1**. (−)-ESI-MS analysis of LOC100003999 and zgc:113054 reaction products.

**Figure supplement 2**. ¹H NMR spectrum of gadusol obtained from *E. coli* cell free extracts containing LOC100003999 and zgc:113054 reactions.

**Figure supplement 3**. Proposed mode of formation of gadusol in vertebrates.

**Figure supplement 4**. The Pentose Phosphate Pathway (*Asamizu et al., 2012*) and the shunt pathway to gadusol.

To test the UV-protective activity of gadusol, a yeast *rad1Δ* mutant, which is sensitive to UVB, was suspended at approximately $10^7$ cells/mL in the concentrated supernatant from the gadusol-producing yeast strain or from an otherwise isogenic control strain that did not produce gadusol. The gadusol-containing supernatant suppressed the UVB-sensitivity of the *rad1Δ* mutant (*Figure 2I*),

confirming the UVB-protective activity of gadusol. Analogous experiments with a wild-type strain (RAD1) at higher doses of UVB showed comparable results (*Figure 2J*), consistent with UVB protective activity.

As noted above, the SPCs EEVS, EVS, DDGS, aminoDHQS, and DOIS are all related to DHQS and are widespread in bacteria and fungi, but other than this report, are not known to exist in vertebrates or prevertebrates. We suggest that the vertebrate EEVS and MT-Ox genes were most plausibly acquired via horizontal gene transfer. Interestingly, searches identify the stramenopile *Aureococcus anophagefferens* and the microalgae *Coccomyxa subellipsoidea*, as the only non-vertebrate organisms in current databases that harbor a similar bifunctional MT-Ox gene, and both organisms have a predicted EEVS gene adjacent to that of MT-Ox. As algae are known to be active horizontal gene transfer agents (*Ni et al., 2012*), algae such as these become a plausible place both for the development of this alternate pathway for gadusol production and as a source of the genes found in vertebrates. Further supporting such a relationship, the *A. anophagefferens* EEVS protein is substantially more similar to the vertebrate EEVSs than it is to bacterial EEVSs (*Figure 1A* and *Figure 1—figure supplement 2*, denoted by the blue star).

Further bioinformatics studies also showed that the tunicates and lancelets lack the EEVS and MT-Ox genes, suggesting that the gene transfer occurred sometime during the evolution of primitive chordates to bony fishes (*Figure 3*). While the EEVS and MT-Ox genes are retained in modern ray-finned fish (with the exception of puffer fish), as well as in amphibians, reptiles, and birds, they were lost in mammals, including the egg-laying mammal platypus, indicating the lack of a direct link between gadusol and the mode of reproduction. The West African coelacanth genome (*Amemiya et al., 2013*) also appears to lack the EEVS and MT-Ox genes (*Figure 3*, *Table 1*). This rare ovoviviparous fish lives in caves 100–500 meters deep and feeds at night, and its lack of gadusol production ability may be directly related to its limited exposure to UV light and/or oxidative stress. Other than the well-documented presence of gadusol in fish eggs, where it can serve to protect the roe from UV damage (*Arbeloa et al., 2010*; *Colleter et al., 2014*), nothing is known of its role(s) in fish, reptiles, amphibians, and birds. Exploring its function in these organisms will increase our understanding of their physiology and ecology.

# Materials and methods

## Molecular phylogenetic analysis

For phylogenetic analysis, full-length amino acid sequences and vertebrate mRNA sequences were analyzed. A reciprocal BLAST hit analysis was performed with the EEVS protein (see *Supplementary file 3*). Sequences were aligned using MUSCLE. ProtTest was used to determine the best model of protein evolution (LG+G) (*Darriba et al., 2011*), and MEGA6 was used to determine the best fit nucleic acid evolutionary model (K80+G) (*Tamura et al., 2013*). RAxML was used for maximum likelihood analysis, and the robustness of the trees was assessed by bootstrap analysis (1000 replicates) (*Stamatakis, 2014*). Bayesian analysis was performed by MrBayes (version 3.2.3), using a random starting tree, running eight chains for 4,000,000 generations, sampling every 250 trees (*Ronquist et al., 2012*). The first 5000 trees were discarded as the burnin, with the remaining trees used to calculate posterior probability. RAxML and MrBayes were run on the CIPRES science gateway (*Miller et al., 2010*). MEGA6 was used for maximum likelihood analysis of vertebrate mRNA sequences with tree robustness assessed by bootstrap (500 replicates). Sources of proteins for the analyses are listed in *Supplementary file 1*.

## Construction of LOC100003999 and zgc:113054 gene expression vectors

The LOC100003999 gene was codon optimized for *E. coli* and synthesized commercially (GeneScript USA Inc., Piscataway, NJ). The product was cloned into *Eco*RV site of pUC57-kan vector. The plasmid was digested with *Bgl*II and *Eco*RI and ligated into *Bam*HI and *Eco*RI site of pRSET-B (Life Technologies, Carlsbad, CA) for the expression of N-terminal hexa-histidine-tagged protein. The zgc: 113054 gene was also codon optimized for *E. coli* and commercially synthesized (GeneScript USA Inc.). The product was cloned into *Eco*RV site of pUC57-amp vector. The plasmid was digested with *Bgl*II and *Eco*RI and ligated into *Bam*HI and *Eco*RI site of pRSET-B (Life Technologies) for the expression of N-terminal hexa-histidine-tagged protein.

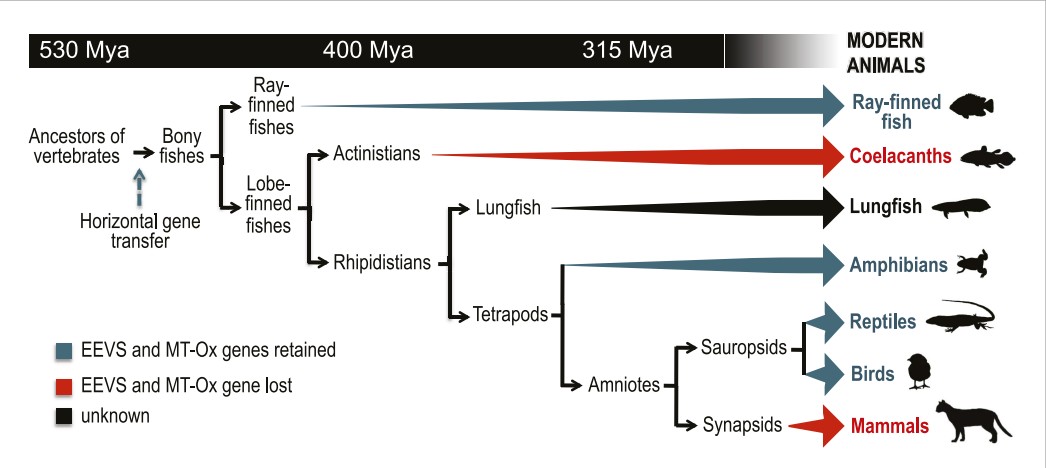

**Figure 3**. Model timeline for the evolution of the EEVS and MT-Ox genes in vertebrates. The genes entered an early vertebrate genome as a linked pair (vertical blue arrow) and were retained in the modern ray-finned fish, amphibians, reptiles, and birds as indicated by thick dark cyan arrows. Coelacanths and mammals lost the genes (thick red arrows). No full genome sequence is available for assessing the presence of EEVS and MT-Ox in lungfish. The phylogenetic trees of the EEVS and MT-Ox proteins or mRNA from a selected set of vertebrates can be found in *Figure 3—figure supplements 1–3* and *Supplementary files 1, 2*.

The following figure supplements are available for figure 3:

**Figure supplement 1**. Phylogenetic tree of the EEVS proteins from a selected set of vertebrates.

**Figure supplement 2**. Phylogenetic tree of the MT-Ox proteins from a selected set of vertebrates.

**Figure supplement 3**. Maximum likelihood tree of vertebrate EEVS mRNA sequences.

## Expression of valA, LOC100003999, and zgc:113054 genes in *E. coli*

pRSETB-valA, pRSETB-LOC100003999, and pRSETB-zgc:113054 plasmids were individually used to transform *E. coli* BL21 GOLD (DE3) pLysS. Transformants were grown overnight at 37°C on LB agar plate containing ampicillin (100 µg/ml) and chloramphenicol (25 µg/ml). A single colony was inoculated into LB medium (2 ml) containing the above antibiotics and cultured at 37°C for 8 hr. The seed culture (1 ml) was transferred into LB medium (100 ml) in a 500-ml flask and grown at 30°C until $OD_{600}$ reached 0.6. Then, the temperature was reduced to 18°C. After 1-hr adaptation, isopropyl-D-1-thiogalactopyranoside (IPTG) (0.1 mM) was added to induce the N-terminal hexa-histidine-tagged proteins. After further growth for 16 hr, the cells were harvested by centrifugation (5000 rpm, 10 min, 4°C), washed twice with cold water, and stored at −80°C until used.

## Purification of recombinant ValA, LOC100003999, and zgc:113054

Cell pellets from a 400-ml culture of *E. coli* BL21 GOLD (DE3) pLysS containing pRSETB-valA, pRSETB-LOC100003999, or pRSETB-zgc:113054 plasmids were resuspended in 20 ml of B buffer (40 mM Tris-HCl, 300 mM NaCl, 10 mM imidazole, pH 7.5). Cells were disrupted by sonication for 1 min (4 times, 2 min interval) at 13 watts on ice (Probe sonicator, Misonix, Farmingdale, NY). 20 ml of lysate was divided into 2-ml tubes and centrifuged (14,500 rpm, 20 min, 4°C). Soluble fractions were collected and transferred into a 50-ml tube. Ni-NTA (QIAGEN, Valencia, CA) resin (5 ml) was applied into 10-ml vol empty column, and the Ni-NTA resin was equilibrated with B buffer (50 ml, 10 CV). About 20 ml of supernatant from cell lysate was applied to the column (flow rate; 0.8 ml/min). The column was then washed with 100 ml (20 CV) of W buffer (40 mM Tris-HCl, 300 mM NaCl, 20 mM imidazole, pH 7.5) at 0.8 ml/min. The hexa-histidine-tagged proteins were eluted by imidazole addition using a gradient mixer containing 100 ml of W buffer and 100 ml of E buffer (40 mM Tris-HCl, 300 mM NaCl, 300 mM imidazole, pH 7.5). The fractions (150 drops or about 5 ml) were collected and checked by SDS-PAGE (Coomassie Blue staining). Fractions containing pure proteins were combined

(25 ml) and dialyzed against 2 l of D buffer (10 mM Tris-HCl, pH 7.5) 3 times (every 3 hr). Dialyzed protein solution was concentrated by ultrafiltration (MWCO 10 K) to 200 μM and flash frozen in liquid $N_2$ prior to storage at −80°C. The yields of the purified proteins were 57 mg/l for ValA, 18 mg/l for LOC100003999, and 79 mg/l for zgc:113054.

### LOC100003999 assay conditions

Each reaction mixture (25 μl) contained Tris-HCl buffer (20 mM, pH 7.5), $NAD^+$ (1 mM), $CoCl_2$ or $ZnSO_4$ (0.1 mM), SH7P (4 mM), and purified enzymes (0.12 mM). The mixture was incubated at 30°C for 2 hr. ValA (instead of LOC100003999) was used as a positive control. No enzyme (buffer only) was used as a negative control.

### Coupled LOC100003999 and zgc:113054 assay conditions

Each reaction mixture (50 μl) contained potassium phosphate buffer (10 mM, pH 7.4), $NAD^+$ (2 mM), $CoCl_2$ (0.2 mM), SH7P (4 mM), and LOC100003999 cell-free extract (20 μl) was incubated at 30°C. After 6 hr, S-adenosylmethionine (5 mM) and purified zgc:113054 (0.1 mM) were added. The mixture was incubated at 30°C for another 6 hr. ValA was used (instead of LOC100003999) as a positive control. Extract of *E. coli* harboring pRSET B empty vector was used as a negative control.

### Zgc:113054 assay using [6,6-$^2H_2$]-EEV as substrate

A reaction mixture (25 μl) containing potassium phosphate buffer (10 mM, pH 7.4), $NAD^+$ (2 mM), $CoCl_2$ (0.2 mM), S-adenosylmethionine (5 mM), [6,6-$^2H_2$]-EEV (4 mM), and purified zgc:113054 (0.1 mM) was incubated at 30°C for 2 hr. Boiled zgc:113054 was used as a negative control.

### TLC analysis of EEV and gadusol

Analytical TLC was performed using silica gel plates (60 Å) with a fluorescent indicator (254 nm), which were visualized with a UV lamp and ceric ammonium molybdate (CAM) or 5% $FeCl_3$ in MeOH-$H_2O$ (1:1) solutions.

### GC-MS analysis of EEV

The enzymatic reaction mixtures were lyophilized, and the products were extracted with MeOH. The MeOH extract was then dried and Tri-Sil HTP (Thermo Scientific, Waltham, MA) (100 μl) was added and left to stand for 20 min. The solvent was removed in a flow of argon gas, and the silylated products were extracted with hexanes (100 μl) and injected into the GC-MS (Hewlett Packard 5890 SERIES II Gas chromatograph).

### Enzymatic synthesis, purification, and analysis of gadusol

Fifty eppendorf tubes containing reaction mixtures (100 μl each), which consist of potassium phosphate buffer (10 mM, pH 7.4), SH7P (5 mM), $NAD^+$ (2 mM), $CoCl_2$ (0.2 mM), and LOC100003999 cell-free extract (40 μl), were incubated at 30°C. After 6 hr, S-adenosylmethionine (5.5 mM) and zgc: 113054 cell-free extracts (30 μl) were added. The reaction mixtures were incubated at 30°C for another 6 hr. The reaction mixtures were quenched with 2 vol of MeOH, left to stand at −20°C for 20 min, then centrifuged at 14,500 rpm for 20 min. The supernatants were pooled and dried under vacuum. The residual water was frozen and lyophilized. The crude sample was dissolved in water (1 ml) and subjected to Sephadex LH-20 column chromatography using phosphate buffer (2.5 mM, pH 7) as an eluant. Fractions containing the product as judged by MS were combined and lyophilized. Furthermore, the product was purified by HPLC (Shimadzu LC-20AD, $C_{18}$ column [YMC], 250 × 10 mm, 4 μm, flow rate 1 ml/min). Solvent system: MeOH—phosphate buffer (5 mM, pH 7), gradient 1–100% of MeOH (0–40 min). Peak at 12.74 min was collected and dried to give gadusol (0.4 mg). $^1H$ NMR (700 MHz, $D_2O$, cryo-probe): δ 4.10 (s, 1H, H-4), 3.71 (d, $J$ = 12 Hz, H-7α), 3.56 (d, $J$ = 12 Hz, H-7β), 3.49 (s, 3H, $OCH_3$), 2.68 (d, $J$ = 17 Hz, H-6α), 2.38 (d, $J$ = 17 Hz, H-6β). HR-MS (ESI-TOF) (*m/z*): $(M+H)^+$ calculated for $C_8H_{13}O_6$, 205.0707; found, 205.0709.

### Zebrafish lines and embryos

Adult wild-type 5D zebrafish were housed at the Sinnhuber Aquatic Research Laboratory on a recirculating system maintained at 28 ± 1°C with a 14 hr light per 10 hr dark schedule. Embryos were collected from group spawns of adult zebrafish as described previously (*Reimers et al., 2006*), and all

experiments were conducted with fertilized embryos according to Oregon State University Institutional Animal Care and Use Protocols. Embryos were staged accordingly as previously described (*Kimmel et al., 1995*) and collected by hand for all experiments. Embryos were reared in media consisting of 15 mM NaCl, 0.5 mM KCl, 1 mM $MgSO_4$, 0.15 mM $KH_2PO_4$, 0.05 mM $Na_2HPO_4$, and 0.7 mM $NaHCO_3$ (*Westerfield, 2000*).

## Polymerase chain reaction

All polymerase chain reaction (PCR) reactions were performed according to manufacturer's specifications. Cycling conditions: 96°C for 3 min, 95°C for 1 min, 65°C for 1 min, and 72°C for 1 min per kB DNA; 35 cycles were used followed by 10 min at 72°C. All PCR products were characterized on an agarose gel. If needed, the PCR product was excised from the gel and purified using the E.Z.N.A. Gel Extraction Kit (Omega Bio-tek, Norcross, GA).

## Quantitative PCR of zebrafish samples

qPCR was performed on a Applied Biosystems StepOnePlus machine. The super mix PerfeCTa SYBR Green FastMix, ROX (Quanta biosciences, Gaithersburg, MD) was used. cDNA (100 ng) from time points at 6, 12, 24, 48, 72, 96, and 120 hpf was used. Super mix (18 µl) was added to bring the final volume to 20 µl. PCR conditions suggested by the supplier were used. For total RNA isolation, 30 embryos were homogenized in RNAzol (Molecular Research Center, Cincinnati, OH); RNA was purified according to the manufacturer's protocol. RNA was quantified by $A_{260/280}$ ratios measured using a SynergyMx microplate reader (Biotek, Winooski, VT) and analyzed with the Gen5 Take3 module. 1 µg of RNA was used for cDNA synthesis. Superscript III First-Strand Synthesis (Life Technologies) and oligo d(T) primers were used to synthesize cDNA from the total RNA.

## Isolation of gadusol from zebrafish

Embryos were collected and euthanized at 72 hpf by induced hypoxia through rapid chilling on ice for 30 min. Embryo media were removed until about 5 ml were left and frozen at −80°C. Embryos were lyophilized overnight. The freeze-dried embryos were then ground with a pestle and mortar under liquid nitrogen. The powder was collected and placed in a pre-weighed glass vial. The mortar was washed with $MeOH-H_2O$ (80:20), and the solvent was added to the powder. The solvent was evaporated, and powder was weighed. The embryo powder was extracted twice with $MeOH-H_2O$ (80:20). The two extracts were combined, dried, and weighed. The extract was suspended in $MeOH-H_2O$ (80:20) (1 ml) and extracted twice with hexanes. The aqueous layer was recovered, dried, and weighed. The extract was suspended in MeOH for analysis by mass spectrometry. The extract was dissolved in phosphate buffer pH 7.0 for identification by HPLC (Shimadzu SPD-20A system, YMC ODS-A column (4.6 id × 250 mm), MeOH—5 mM phosphate buffer (1% MeOH for 20 min followed by a gradient from 1 to 95% MeOH in 20 min), flow rate 0.3 ml/min, 296 nm. The isolated gadusol was analyzed by MS (ThermoFinnigan LCQ Advantage system) and NMR (in $D_2O$; Bruker Unity 300 [300.15 MHz] spectrometer).

## Construction of yeast mutants

The yeast strains used are listed in *Supplementary file 4*. The *TRP1* gene was replaced in BY4742 *tal1∆:: KanMX4* with a wild-type *URA3* allele from S288c by standard methods (*Baudin et al., 1993*). The deletion was confirmed by PCR using primer pairs TRP1DisUP/TRP1DisLO and URA3DisUP/TRP1DisLO. The BY4742 *tal1∆::KanMX4 trp1∆::URA3* strain was then co-transformed (*Gietz et al., 1992*) with pXP416 and pXP420 to generate an empty vector control strain and with pXP420-EEVS and pXP416-MT-Ox to generate a gadusol-producing strain. The *RAD1* gene was replaced in BY4742 *tal1∆::KanMX4 trp1∆::URA3* with a wild-type *LEU2* allele from S288c by standard methods (*Reynolds et al., 1987*). The deletion was confirmed by PCR using primer pairs RAD1UP/RAD1LO. The resultant BY4742 *tal1∆:: KanMX4 trp1∆::URA3 rad1∆::LEU2* strain was then co-transformed with pXP416 and pXP420.

## Media and yeast growth conditions

Cells were pre-grown in YEPD (1% yeast extract, 2% peptone, and 2% glucose) for transformations, and in YNB (Bacto yeast nitrogen base without amino acids) + 2% glucose supplemented with 30 µg/ml leucine and 30 µg/ml lysine to select for transformants and to produce gadusol. Liquid media were sterilized by filtration using a 0.45-µm filter, and agar-based media were sterilized by autoclaving. Liquid cultures were grown at 30°C for 48 hr and 200 rpm; plates were incubated at 30°C.

## Yeast overexpression plasmid construction

Plasmids are listed in *Supplementary file 5*. Primers used for PCR are listed in *Supplementary file 6*. PCR amplicons with *Spe*I and *Xho*I terminal restriction sites were generated for the EEVS gene and MT-Ox gene using pRSETB-EEVS and pRSETB-MTOx as templates, respectively. The EEVS and MT-Ox amplicons were then digested with *Spe*I and *Xho*I and ligated into *Spe*I-digested pXP420 and *Xho*I-digested pXP420 and pXP416, respectively, and introduced into competent *E. coli* (Top 10; Life Technologies) by transformation. *E. coli* transformants were selected on LB plates supplemented with ampicillin (100 µg/ml). Transformants were then screened by digesting plasmid DNA with *Spe*I and *Xho*I restriction enzymes and analyzing fragments by agarose gel electrophoresis.

## Identification of gadusol production in *S. cerevisiae*

*S. cerevisiae* cell pellets from 5 ml cultures were extracted with MeOH, and the supernatant was extracted with nBuOH. Extracts were concentrated and analyzed by HPLC (Shimadzu SPD-20A system, YMC ODS-A column [4.6 id × 250 mm], MeOH—5 mM phosphate buffer (1% MeOH for 20 min followed by a gradient from 1 to 95% MeOH in 20 min), flow rate 0.3 ml/min, 296 nm.

## Irradiation protocol

A *rad1Δ* mutant (*MATα his3Δ1 leu2Δ0 lys2Δ0 trp1Δ::URA3 ura3Δ0 rad1Δ::LEU2 tal1Δ::KanMX4/* pXP416, pXP420) or wild-type *RAD1* strain (S288c, *MATα SUC2 gal2 mal2 mel flo1 flo8-1 hap1 ho bio1 bio6*) was grown at 30°C and 200 rpm in YNB + 2% glucose + 30 µg/ml leu + 30 µg/ml lys. Cells were harvested after 24 hr by centrifugation, washed twice in the ninefold concentrated supernatant of either the gadusol-producing strain BY4742 *tal1Δ trp1Δ*/pXP416-MTOx, pXP420-EEVS or of the control strain BY4742 *tal1Δ trp1Δ*/pXP416, pXP420, and suspended in the respective concentrated supernatants at $10^7$ cells/ml. Cells (375 µl) were irradiated with UVB (302 nm) at the indicated doses in wells of a 24-well microtiter plate shaken at 900 rpm. 3 µl aliquots of cells were then spotted onto a YEPD plate, which was incubated 24 hr at 30°C prior to being photographed. The supernatants of the gadusol producing and control strains were obtained by centrifugation following 5 day of growth in YNB + 2% glucose + 30 µg/ml leucine + 30 µg/ml lysine at 30°C and 200 rpm. Supernatants were freeze-dried, dissolved in a volume of distilled water 1/10 of the initial culture volume, and stored at 4°C until use. Just prior to suspension of cells, the concentrated supernatant was adjusted to 50 mM phosphate, pH 7.0 resulting in a final ninefold concentrate.

## Acknowledgements

This paper is dedicated to Professor Heinz G Floss on the occasion of his 80th birthday. The authors thank Canan Schumann, Oleh Taratula, Jun Ding, Allen S Yoshinaga, Zachary Landry, Edward Davis, and Seika Mahmud for technical assistance and Sinnhuber Aquatic Research Laboratory for providing zebrafish embryos. This research was supported by the Oregon State University College of Pharmacy General Research Funds and NIH grant P30 ES000210.

## Additional information

### Funding

| Funder | Grant reference | Author |
| --- | --- | --- |
| National Institutes of Health (NIH) | P30 ES000210 | Robert L Tanguay |
| Oregon State University College of Pharmacy | General Research Funds | Taifo Mahmud |

The funders had no role in study design, data collection and interpretation, or the decision to submit the work for publication.

### Author contributions

ARO, KHA, GH, JLD, KMK, Acquisition of data, Analysis and interpretation of data; SA, Conception and design, Acquisition of data, Analysis and interpretation of data; PAK, Analysis and interpretation of data, Drafting or revising the article; RLT, Conception and design, Analysis and interpretation of data; ATB, TM, Conception and design, Analysis and interpretation of data, Drafting or revising the article

**Author ORCIDs**
Taifo Mahmud, http://orcid.org/0000-0001-9639-526X

**Ethics**
Animal experimentation: All procedures involving animals were performed in accordance with the guidelines of the National Institutes of Health. The protocol used (4321) was approved by the Oregon State University Institutional Animal Care and Use Committee.

## Additional files

**Supplementary files**
• Supplementary file 1. SPCs used for multiple sequence alignment and phylogenetic tree construction.

• Supplementary file 2. MT-Ox proteins used for multiple sequence alignment and phylogenetic tree construction.

• Supplementary file 3. Reciprocal best hit analysis.

• Supplementary file 4. Yeast strains used.

• Supplementary file 5. Plasmids used.

• Supplementary file 6. Primers used.

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
