## [Decision Letter]

Thank you for sending your work entitled “De novo Synthesis of a Sunscreen Compound in Vertebrates” for consideration at *eLife*. Your article has been favorably evaluated by Ian Baldwin (Senior Editor) and 3 reviewers, one of whom is a memaber of our Board of Reviewing Editors.

The following individuals responsible for the peer review of your submission have agreed to reveal their identity: Reviewing editor, Wilfred van der Donk; Emily Balskus, peer reviewer. A third reviewer remains anonymous.

The Reviewing editor and the other reviewers discussed their comments, and all three reviewers agree that this study is of general interest to the *eLife* readership, provided the authors can address the concerns listed below.

The authors show in this manuscript that genes encoding a 2-epi-5-epi-valiolone synthase (EEVS) and a bifunctional O-methyltransferase/NAD+ dependent oxidoreductase genes are present in various egg-laying vertebrates. They also show that the two genes are expressed in zebrafish and that gadusol can be isolated from zebrafish embryos. The authors introduced both genes into an engineered strain of yeast that accumulates the EEVS substrate sedoheptulose-7-phosphate and demonstrate the production of gadusol. Finally, they show that extracts from gadusol-producing yeast protect both wild-type and mutant yeast strains from UV-B radiation. Collectively, these data provide strong support that vertebrates can make these compounds and that they need not be produced by symbionts or obtained in diet. Overall, the biochemical data is very convincing, with some questions raised by the reviewers. The bioinformatics/evolution data was considered less robust and needs strengthening. Results from the phylogenetic analyses suggest a series of repeated transfers between bacterial and eukaryotic homologs that appear to be an interesting part of the story. Yet, the methods listed are not sufficient to make this claim. Below are several ways to validate/improve these phylogenetic results. The reviewers agree that the requested revisions should be doable.

The Reviewing editor has assembled the following comments to help you prepare a revised submission:

1) The inclusion of bacterial and eukaryotic homologs into the evolutionary history of these genes needs to be justified by a reciprocal blast hit analysis in which the authors switch the BLAST query sequence between bacteria and eukaryotes to see if they always retrieve the same hits in the database. This concern is particularly relevant when using protein sequences, because convergent evolution rather than horizontal gene transfer may drive the appearance of reciprocal orthology when there isn't any.

2) A model of protein evolution needs to be statistically selected to build the tree. The authors appear to have used the simple, default parameter in Geneious, which is insufficient for accuracy in phylogenetic tree building. To select a model of protein evolution, ProtTest needs to be run, and the selected model needs to be implemented in the tree building.

3) In addition to building the tree with protein sequences, can the authors build the tree with DNA sequences? The DNA divergences may be too high to do this for all of the sequences in the tree, but subsections of the tree could be reanalyzed with DNA sequences, starting with reciprocal blast analyses and then model selection (using ModelTest for DNA) and tree building.

4) It is highly desirable to use multiple phylogenetic inference methods to validate evolutionary relationships. Thus, in addition to the reported Max Likelihood analysis, a Mr. Bayes analysis should be run alongside.

5) Bootstrap and posterior probability support values must be reported in a phylogenetic tree analysis to evaluate the strength of the evolutionary results.

6) The authors express and purify the proteins of interest, but the experimental section suggests that for some experiments they were assayed as cell free extracts. It is not clear which results discussed in the text and shown in the figures were obtained with purified protein and which with cell free extracts, but the reader is left with the impression that all data was obtained with purified proteins. The authors should also explain why cell free extracts were used when purified proteins were apparently available.

7) The authors should provide more information about the in vitro biochemical characterization of the zebrafish EEVS. Did they test the activity of the enzyme toward sugar substrates other than sedoheltulose-7-phosphate? Also the TLC shown in Figure 1—figure supplement 2 seems to show multiple new products being formed by this enzyme. These products do not appear to be present in the assay with ValA, the bacterial EEVS used as a positive control. What are these alternate products?

8) The authors should provide some additional commentary in their discussion regarding the proposed biological roles of gadusol and MAAs in fish and the issue of their origin (endogenous synthesis vs dietary), and highlight issues that remain unresolved with the current results. For instance, it has been demonstrated experimentally that fish can accumulate MAAs from their diets (see Mar Biol 144:1057-1064, Comp Biochem Physiol Part A Mol Integr Physiol 120:587-598). Are the relative concentrations of gadusol and MAAs in fish roe known? Just because there is an endogenous pathway for gadusol synthesis in fish does not mean that this compound cannot also be acquired from dietary sources along with the MAAs or that it is the primary contributor to UV protection.

---

## [Author Response]

*1) The inclusion of bacterial and eukaryotic homologs into the evolutionary history of these genes needs to be justified by a reciprocal blast hit analysis in which the authors switch the BLAST query sequence between bacteria and eukaryotes to see if they always retrieve the same hits in the database. This concern is particularly relevant when using protein sequences, because convergent evolution rather than horizontal gene transfer may drive the appearance of reciprocal orthology when there isn't any*.

We have performed a reciprocal blast hit analysis by switching the BLAST query sequence between bacteria and eukaryotes and found that from 55 bacterial EEVS sequences blasted against the vertebrate EEVS sequences, 17 unique vertebrate EEVS subjects were hit. Then the vertebrate EEVS sequences were blasted against all proteins (minus vertebrate proteins) used for phylogenetic analysis, which hit 7 unique bacterial EEVS subjects. Two pairs of reciprocal best hits were identified; *Actinoplanes* sp. A40644 (BAD07382.1) was best hit with *Falco peregrinus* (XP_005230087.1), and *Cystobacter fuscus* DSM 2262 (EPX59479.1) with *Dicentrarchus labrax* (CBN80976.1). The result is presented in [Supplementary-material SD4-data].

*2) A model of protein evolution needs to be statistically selected to build the tree. The authors appear to have used the simple, default parameter in Geneious, which is insufficient for accuracy in phylogenetic tree building. To select a model of protein evolution, ProtTest needs to be run, and the selected model needs to be implemented in the tree building*.

As suggested, we have used ProtTest to determine the best model of protein evolution (LG+G).

*3) In addition to building the tree with protein sequences, can the authors build the tree with DNA sequences? The DNA divergences may be too high to do this for all of the sequences in the tree, but subsections of the tree could be reanalyzed with DNA sequences, starting with reciprocal blast analyses and then model selection (using ModelTest for DNA) and tree building*.

We did try to build the tree with DNA sequences, and as suspected by the reviewers the DNA divergences are too high and we didn’t see a clear separation between clades. We then decided to focus only on the EEVS DNA sequences, but still didn’t see a clear separation between bacterial and vertebrate sequences. Finally, we narrowed down our analysis to vertebrate mRNAs only, and the result is shown in Figure 3—figure supplement 3. MEGA6 was used to determine the best fit nucleic acid evolutionary model (K80+G), and for maximum likelihood analysis of the vertebrate mRNA sequences with tree robustness assessed by bootstrap (500 replicates).

*4) It is highly desirable to use multiple phylogenetic inference methods to validate evolutionary relationships. Thus, in addition to the reported Max Likelihood analysis, a Mr. Bayes analysis should be run alongside*.

We have used RAxML for maximum likelihood analysis, and the robustness of the trees was assessed by bootstrap analysis (1000 replicates). As suggested, Bayesian analysis was performed by MrBayes (version 3.2.3), using a random starting tree, running eight chains for 4,000,000 generations, sampling every 250 trees. The first 5,000 trees were discarded as the burnin, with the remaining trees used to calculate posterior probability. RAxML and MrBayes were run on the CIPRES science gateway. The Bayesian phylogenetic tree is presented in Figure 1, whereas the maximum likelihood phylogenetic tree is presented in Figure 1—figure supplement 2. The results are fully consistent with the previous ones, and even place both algal EEVS sequences closer to the vertebrate sequences, further strengthening our horizontal gene transfer proposal.

*5) Bootstrap and posterior probability support values must be reported in a phylogenetic tree analysis to evaluate the strength of the evolutionary results*.

We have included the bootstrap and posterior probability values for main branches in the phylogenetic trees. For clarity, values for branches within the clades were excluded.

*6) The authors express and purify the proteins of interest, but the experimental section suggests that for some experiments they were assayed as cell free extracts. It is not clear which results discussed in the text and shown in the figures were obtained with purified protein and which with cell free extracts, but the reader is left with the impression that all data was obtained with purified proteins. The authors should also explain why cell free extracts were used when purified proteins were apparently available*.

Characterization of the zebrafish EEVS was carried out with a purified LOC100003999 protein as shown in Figure 1—figure supplement 3. However we discovered that the purified EEVS was relatively unstable and the activity was much lower than that in the cell free extracts. Therefore, for the characterization of the second enzyme (purified MT-Ox), we used cell free extracts containing EEVS to make sure that enough EEV (the substrate for MT-Ox) is produced. We have specified whether purified proteins or cell-free extracts were used in the figure legends.

*7) The authors should provide more information about the in vitro biochemical characterization of the zebrafish EEVS. Did they test the activity of the enzyme toward sugar substrates other than sedoheltulose-7-phosphate? Also the TLC shown in*
Figure 1—figure supplement 2
*seems to show multiple new products being formed by this enzyme. These products do not appear to be present in the assay with ValA, the bacterial EEVS used as a positive control. What are these alternate products?*

Since the amino acid sequence of zebrafish EEVS is highly similar to the bacterial EEVS, sharing all the 14 conserved active site residues, we only tested the activity of the protein toward sedoheptulose 7-phosphate and confirmed its identity as an EEVS. We did not test the activity of the enzyme toward other sugar substrates. Regarding the additional spots on the TLC, we are not sure what they are. However, they do not appear to be the products of the enzyme. We have replaced the TLC with a new one with no or very little of these spots.

*8) The authors should provide some additional commentary in their discussion regarding the proposed biological roles of gadusol and MAAs in fish and the issue of their origin (endogenous synthesis vs dietary), and highlight issues that remain unresolved with the current results. For instance, it has been demonstrated experimentally that fish can accumulate MAAs from their diets (see Mar Biol 144:1057-1064, Comp Biochem Physiol Part A Mol Integr Physiol 120:587-598). Are the relative concentrations of gadusol and MAAs in fish roe known? Just because there is an endogenous pathway for gadusol synthesis in fish does not mean that this compound cannot also be acquired from dietary sources along with the MAAs or that it is the primary contributor to UV protection*.

Based on the present study, we believe that gadusol and MAAs found in marine fish are synthesized through two different pathways. While we propose that gadusol is synthesized de novo by the fish, the accumulation of MAAs in fish would still appear to be of dietary origin. Therefore, we have added the following comment in the Discussion “However, as MAAs are synthesized via a different pathway and there is no evidence that fish have those biosynthetic enzymes, the accumulation of MAAs in fish would still appear to be of dietary origin (24; 46).”